# ROBUST POLICY OPTIMIZATION WITH EVOLUTIONARY TECHNIQUES

## ABSTRACT

Learning-based techniques to train control policies of autonomous agents often assume that the agent experiences are sampled according to a certain dynamical model for the environment. However, environmental dynamics can change (due to intentional or unintended changes to the environment). While domain randomization and robust learning can handle some distribution shifts, large environmental shifts may necessitate re-training to learn policies optimal in the changed environment. We present an algorithm called 'Evolutionary Robust Policy Optimization' (ERPO) inspired by evolutionary game theory (EGT) to address the problem of incrementally and efficiently adapting policies to an altered environment. We give theoretical guarantees on the convergence of our algorithm to the optimal policy under the assumption of sparse rewards. We empirically demonstrate that our algorithm outperforms several state-of-the-art deep RL algorithms in many gym environments. Specifically, we are able to adapt policies using fewer training steps while getting higher rewards and requiring lower overall computation time.

## 1 INTRODUCTION

Large-scale deployment of autonomous systems is fast becoming a reality with applications in autonomous driving, automated warehouses, and multi-UAV missions (Paul & Deshmukh, 2022). One of the main challenges for such systems is planning, i.e., for each state the agent may be in, deciding which action it should take. There are several computationally efficient approaches for planning the agent's actions in deterministic and stochastic environments, especially when a model of the environment is available (Luna et al., 2022; Elmaliach et al., 2009) (Varambally et al., 2022; Bellusci et al., 2020) (Li et al., 2021b) (Li et al., 2021a; Morris et al., 2016). However, many autonomous systems are now deployed in highly uncertain and dynamic environments, where such a model may not be available. Some reinforcement learning (RL) algorithms (Sutton & Barto, 2018) (Bertsekas, 2019) have been highly effective at learning optimal policies even when the environment dynamics are unknown. Deep RL algorithms (Lowe et al., 2017; Kuznetsov et al., 2020; Haarnoja et al., 2018) have enabled learning policies in continuous state environments with continuous actions enabling applications in robot control (Lillicrap et al., 2019; Hessel et al., 2017).

A prevalent concern is the propensity of these algorithms to overfit a model, thus losing robustness and generalizability to changes in the environment. Some efforts have sought to fortify these algorithms through techniques such as domain randomization (Peng et al., 2018) and distributionally robust reinforcement learning techniques (Smirnova et al., 2019) (Pinto et al., 2017). Current approaches in Robust RL (Pinto et al., 2017) focus on enabling model adaptation to bridge the gap between simulation and real-world applications. Simulation models are generally not very realistic, and fail to consider environmental variables such as resistance, friction, and various other minor disturbances, (commonly referred to as the Sim2Real gap) so they cannot be directly deployed in such applications. There is also work in adversarial RL (Zhang et al., 2021) that focuses on optimizing the extent of noise in the environment to train the model to be robust in the worst case, or even training in the presence of adversarial actions (Tessler et al., 2019). There is also theoretical work in developing versions of DQN such as DQN-Uncertain Robust Bellman Equation (Derman et al., 2020) that focuses on developing Robust Markov Decision Processes (RMDPs) with a Bayesian approach.

Methods such as proximal policy optimization with domain randomization (PPO-DR) achieve better performance than traditional deep RL, and algorithms such as Monotonic Robust Policy Optimization (Jiang et al., 2021) provide lower bounds for the worst-case performance of a given policy. Control theory approaches (Rajeswaran et al., 2016) train over under-performing subsets of trajectories and yield policies that demonstrate greater resilience in worst-case scenarios.

Despite these advances, robust adaptation of pre-trained models to environments with substantial changes is a sizeable challenge. For example, consider situations where the factory floor layout is altered, driving paths in a warehouse are blocked due to debris or other temporary obstacles, or road network topology is affected due to a natural disaster, accidents or construction. Each of these situations represents a significant distribution shift in the dynamics: specific actions previously enabled in the training environment may no longer be available or be severely sub-optimal in the new environment. Many such situations often necessitate either complete retraining or, at the least, extensive adjustments to hyper-parameters alongside substantial training efforts.

## 1.1 Contributions:

To address this limitation, we propose a novel approach that couples RL-based planning with evolutionary principles. Drawing inspiration from the 'vine' strategy seen in Trust Region Policy Optimization (TRPO) (Schulman et al., 2017a), we generate batches of trajectories using a simulator for the perturbed environment. In this new environment, we explore by employing an $\epsilon$-soft version of the optimal policy from the original environment. We then use this data to improve the existing policy by weighting state-action pairs that exhibit high fitness or returns in trajectories. The incremental modification of the policy with each new batch of data obviates the use of gradients. It is instead inspired by the evolutionary game-theoretic idea of *replicator dynamics*. We establish theoretical guarantees that our algorithm attains convergence to optimality.

We benchmark our algorithm against the prevailing mainstream deep-RL methodologies, including PPO (Schulman et al., 2017b), PPO-DR (Peng et al., 2018), DQN (Mnih et al., 2013), and A2C (Mnih et al., 2016) trained from scratch on the new model and trained from the base model of the unperturbed environment (denoted as PPO-B, DQN-B and A2C-B respectively). Our approach outperforms these methods in many standard gym environments used in RL settings. As our work is currently limited to discrete state and action spaces, we use larger and more complex versions of *FrozenLake*, *Taxi*, *CliffWalking*, *Minigrid: DistributionShift* environment and a *Minigrid* environment with walls and lava (*Walls&Lava*). Specifically, we show that we have shorter computation times and fewer training episodes required to achieve comparable performance levels.

## 2 Preliminaries

We model the system consisting of an autonomous agent interacting with its environment as a Markov Decision Process (MDP) defined as the tuple: $(S, A, \Delta, R, \gamma, \mathcal{I})$, where, $S$ denotes the set of possible states of the agent. At each time-step $t$, we assume that it is in some state $s_t$ and executes action $a_t$ transitioning to the next state $s_{t+1}$ and receiving a reward $r_{t+1}$. The transition dynamics $\Delta$ is a probability distribution over $S \times A \times R \times S$. In other words, $\Delta$ is the distribution of MDP transitions or tuples $(s_t, a_t, r_{t+1}, s_{t+1})$. It is often convenient to abuse notation, and use $(s_{t+1}, r_{t+1}) \sim \Delta(s_t, a_t)$ to denote a transition sampled from $\Delta$, where the first two elements of the tuple are $s_t, a_t$ respectively.

We assume that we are provided with a set of initial states $\mathcal{I} \subseteq S$. At time 0, we assume that an agent is assigned an initial state sampled randomly from $\mathcal{I}$, i.e., $s_0 \sim \mathcal{I}$. We assume that the *control policy* is a *stochastic* policy $\pi$, i.e., at any time $t$, with the agent in state $s_t$, the action $a_t$ is sampled from the distribution $\pi(a \mid s = s_t)$.

We are interested in optimizing policies over a finite time horizon $T$. A trajectory $\tau^\pi$ of the agent induced by policy $\pi$ is defined as a $(T+1)$-length sequence of state-action pairs:

$$\tau = \{(s_0, a_0), (s_1, a_1), \ldots, (s_T, a_T)\}, \text{where,} \quad \begin{aligned} &\forall i < T : a_i \sim \pi(a \mid s = s_i), \text{ and,} \\ &\forall i < T : (s_i, a_i, r_{i+1}, s_{i+1}) \sim \Delta. \end{aligned} \quad (1)$$

The return function $G_\Delta^\pi$ maps a given trajectory $\tau^\pi$ (induced by the policy $\pi$ under the transition dynamics $\Delta$) to some number in $\mathbb{R}$. In particular, several RL papers use the discounted sum of rewards accumulated over the trajectory defined as $G_\Delta^\pi(\tau) = \sum_{i=0}^{T-1} \gamma^i r_{i+1}$, where, we assume that for every

state-action pair $(s_i, a_i) \in \tau$, the reward $r_{i+1}$ corresponds to the transition $(s_i, a_i, r_{i+1}, s_{i+1}) \sim \Delta$. We can then define the *value*, and the *action-value* of a state, and the expected return, under a given policy $\pi$ and transition dynamics $\Delta$ as follows:

$$v_\Delta^\pi(s_t) = \mathbb{E}_{\substack{(s_{t+i+1}, r_{t+i+1}) \sim \\ \Delta(s_{t+i}, a_{t+i})}} \left[ \sum_{i=0}^\infty \gamma^i r_{t+i+1} \right]; \quad Q_\Delta^\pi(s_t, a_t) = \mathbb{E}_{\substack{(s_{t+i+1}, r_{t+i+1}) \sim \\ \Delta(s_{t+i}, a_{t+i})}} \left[ \sum_{i=0}^\infty \gamma^i r_{t+i+1} \right] \quad (2)$$

Here, we use $\tau \sim \Delta$ to indicate that $\tau$ is sequence of state-action pairs as defined in equation 1, and for each consecutive $(s_i, a_i), (s_{i+1}, a_{i+1}), (s_i, a_i, r_{i+1}, s_{i+1}) \sim \Delta$. The expected discounted return for an agent under the policy $\pi$ (and with the transition dynamics $\Delta$) across all initial states of the MDP can then be defined Eq. equation 3

$$\eta_\Delta(\pi) = \mathbb{E}_{s_0 \sim \mathcal{I}} [G_\Delta^\pi(\tau)] = \mathbb{E}_{\substack{s_0 \sim \mathcal{I} \\ (r_{i+1}, s_{i+1}) \sim \Delta(s_i, a_i)}} \left[ \sum_{i=0}^{T-1} \gamma^i r_{i+1} \right] \quad (3)$$

An optimal policy $\pi_\Delta^\star$ for the MDP with transition dynamics $\Delta$ is then defined as the one that maximizes $\eta_\Delta(\pi)$, i.e.,

$$\pi_\Delta^\star = \arg\max_\pi \eta_\Delta(\pi)$$

**Problem Definition.** Suppose that there is a significant perturbation in the distribution representing the environment dynamics. Let the new environment dynamics be denoted by $\Delta_{new}$. The problem we wish to solve is to learn a new policy $\pi_{\Delta_{new}}^\star$, s.t.,

$$\pi_{\Delta_{new}}^\star = \arg\max_\pi \eta_{\Delta_{new}}(\pi) \quad (4)$$

*Assumptions:*

1. We assume we have a procedure that yields us $\pi_\Delta^\star = \arg\max_\pi \eta_\Delta(\pi)$. (From here on, we will simply refer to $\pi_\Delta^\star$ as $\pi_{old}^\star$ or $\pi^*$.
2. Let $\beta = D_{TV}(\Delta || \Delta_{new})$, i.e. the total variation distance between the transition dynamics of the old and new environments; then $\beta$ is bounded as $\beta \leq \beta_{hi}$.
3. The expected performance of the previously optimal policy in the new environment is bounded as:

$$\eta_\Delta(\pi_{old}^\star) - C \leq \eta_{\Delta_{new}}(\pi_{old}^\star) \leq \eta_\Delta(\pi_{old}^\star) \quad (5)$$

where C is a non-negative constant, $C \geq 0$ which indicates that the old optimal policy cannot yield better expected returns in the new environment as opposed to the old one. (This has been proven in (Jiang et al., 2021).)

## 3 SOLUTION

We use the principles of evolutionary game theory (EGT) (Smith, 1982), (Sandholm, 2009) to approach the solution to this problem. We define a fitness function over the trajectory induced by the policy. The fitness can be synonymous with the trajectory's return, i.e. $f(\tau|\pi) = G^\pi(\tau|p)$.[1] A commonly used dynamic equation in Evolutionary Game Theory is the *replicator equation* (Sandholm, 2009), which for our scenario can be represented as:

$$\pi^{i+1} = \pi^i \times [f(\tau|\pi) - \sigma(\pi)] \ where \ \sigma(\pi) = \eta_\Delta(\pi) \quad (6)$$

i.e., the policy in the $(i+1)^{th}$ iteration is given by the product of the policy in the $i^{th}$ iteration and the difference between the fitness of the trajectory and the average fitness of the population. In our scenario, the average fitness is the same as the expected sum of discounted rewards. This equation ensures that the policy is incremented if the fitness of a trajectory is greater than the average fitness and vice versa.

Let us use the notation $\tau_s$ to denote the set of all trajectories that contain $s$, and similarly use $\tau_{(s,a)}$ to indicate the set of trajectories containing the pair $(s, a)$.

As we make state-wise updates, we modify the replicator equation to be:

$$\pi^{i+1} = \pi^i \cdot \alpha \cdot \frac{\mathbb{E}[f(\tau_{s,a})|\pi]}{\mathbb{E}[f(\tau_s)|\pi]} \quad (7)$$

---

[1]From here on, we use the fitness notation in place of the return

---

**Algorithm 1:** EVOLUTIONARY ROBUST POLICY OPTIMIZATION

---

**Input** :
- $\pi^\star = \arg\max_\pi \eta_\Delta(\pi)$
- $\forall s \in S, a \in A : \pi^0_{new}(s,a) = 1/|A|$
- $\epsilon, \nu \in (0,1), \delta > 0;$ and $\alpha > 1$

**Output:** $\pi^\star_{new} = \arg\max_\pi \ \eta_{\Delta_{new}}(\pi)$

**1** $i \leftarrow 0;\ \eta^0 \leftarrow \eta_{\Delta_{new}}(\pi^\star)\ \pi^0_{train} \leftarrow w\pi^\star + (1-w)\pi^0_{new}$ **do**
**2**    **for** $b = 1$ *to Number of Episodes per Batch* **do**
**3**      Sample the initial state: $s_0 \sim \mathcal{I}$
**4**      **for** $t = 1\ to\ T$ **do**
**5**        Sample action as per training policy: $a_t \sim \pi_{train}$
**6**        Sample next state: $(s_t, a_t) \sim \Delta_{new}$
**7**        Append the transition to the trajectory $\tau_b = \tau_b \circ (s_t, a_t, s_{t+1})$
**8**      Append trajectory $\tau_b$ to batch
**9**    **for** *all trajectories $\tau_b$ in batch* **do**
**10**      **for** *each $(s,a) \in \tau_b$* **do**
**11**        Update $\pi^{i+1}_{new}(s,a) \leftarrow \pi^i_{new}(s,a) \cdot \alpha \cdot \frac{\mathbb{E}[f(\tau_{s,a})]}{\mathbb{E}[f(\tau_s)]}$
**12**        Normalize $\pi^{i+1}_{new}$
**13**    $\eta^{i+1} \leftarrow \eta_{\Delta_{new}}(\pi^i_{train}); \pi^{i+1}_{train} \leftarrow w\pi^\star + (1-w)\pi^{i+1}_{new}$
**14**    $w \leftarrow w - \nu; i \leftarrow i + 1$
**15** **while** $( (w > \epsilon)\ or\ (\eta^{i+1} - \eta^i > \delta) )$
**16** Output $\pi_{train}$ as $\pi*_{new}$ .

---

where $\alpha \in \mathbb{R}^+$ and $\alpha < 1$, and $(s, a, \cdot, s') \sim \Delta$. i.e., we use the ratio of the expected value of fitness of all trajectories containing the pair $(s, a)$ to the expected value of the fitness of all the trajectories containing $s'$ rather than the difference. We also use a constant $\alpha$, which allows us to control the rate of change of the population depending on our environment rewards.

### 3.1 PROPOSED ALGORITHM

Our algorithm uses batch-based updates: We initialize our training policy to be a weighted combination of the old optimal policy $\pi^\star$, and our new policy $\pi_{new}$- which is initially random (see 1). We sample trajectories as part of a batch, and the state-action pairs in these trajectories are updated according to the update rule 11. The return for the $(i + 1)^{th}$ iteration is set as the return under the training policy, and the training policy is updated 13, to take into account the update to $\pi_{new}$ in 11. The weight assigned to the old policy is decremented with each iteration 14. This process is repeated until both our termination conditions have been met 15, i.e.

1. $(\eta^{i+1} - \eta^i > \delta)$ ): This condition checks if our policy is improving with each update. If it does not, we say it has converged.

2. $w > \epsilon$: As $w$ is the weight assigned to $\pi^*$, we decrement it by some $0 < \nu < 1$ with each batch update, until $w \leq \epsilon$, which will indicate that our policy $\pi_{train}$, is an $\epsilon-$soft policy.

### 3.2 PROOF OF CONVERGENCE:

We first provide a list of assumptions required for our proposed algorithm to converge.

1. We assume the existence of at least one feasible solution in the new environment.
2. We assume a *sparse reward* setting, where very few (or only the goal states) have an informative reward. In such cases, the value of a state can be approximated to the average return across all trajectories containing the state.
3. Additionally, for the sake of the proof, we assume that the rewards are non-negative. The results in (Ng et al., 1999) show that policies are invariant to reward scaling, which allows us to use this algorithm even in scenarios with negative rewards.

**Theorem 1.** *The algorithm ERPO converges to an optimal policy in the new environment, i.e.:*

$$\pi^*_{new} \;=\; \arg\max_{\pi} \eta_{\Delta_{new}}(\pi) \tag{8}$$

*Proof.* As our system is an MDP, we know that at least one optimal policy must exist (Sutton & Barto, 2018). Our algorithm ERPO is based on policy iteration so to show convergence, we show that $v_{\pi^i}(s) \le v_{\pi^{i+1}}(s)$, where $\pi^i$ and $\pi^{i+1}$ are the policies in the $i^{th}$ and $(i+1)^{th}$ iterations respectively. Therefore, we show that our value function monotonically improves with each algorithm iteration to prove convergence.

We partition the action space $A$ into the following sets:

- $A_h(s) = \{a \in A | v_{\pi^i}(s') \ge \mathbb{E}[v_{\pi^i}(s)], \text{ where } (s, a, \cdot, s') \sim \Delta\}$
- $A_l(s) = \{a \in A | v_{\pi^i}(s') < \mathbb{E}[v_{\pi^i}(s)], \text{ where } (s, a, \cdot, s') \sim \Delta\}$

Now we can denote $v_{\pi^i}(s)$ as:

$$v_{\pi^i}(s) = \sum_{a_h \in A_h(s)} \pi^i(a_h|s)Q_{\pi^i}(s, a_h) \;+\; \sum_{a_l \in A_l(s)} \pi^i(a_l|s)Q_{\pi^i}(s, a_l) \tag{9}$$

According to our update rule:

$$\pi^{i+1}(s, a) \;=\; \pi^i(s, a) \times \frac{\mathbb{E}[f(\tau_{(s,a)})]}{\mathbb{E}[f(\tau_s)]}$$

By our sparse reward assumption 2, we can say that $v(s') \approx \mathbb{E}[f(\tau_s)]$, and similarly, $Q(s, a) \approx \mathbb{E}[f(\tau_{(s,a)})]$ and so:

$$\pi^{i+1}(s, a) \;=\; \pi^i(s, a) \times \frac{Q_{\pi^i}(s, a)}{\mathbb{E}[v_{\pi^i}(s)]}$$

By our assumption of non-negative rewards, 3, $\because v_{\pi^i}(s'_h) \ge \mathbb{E}[v_{\pi^i}(s)], Q_{\pi^i}(s, a_h) \ge \mathbb{E}[v_{\pi^i}(s)]$

$$\therefore \pi^{i+1}(s, a_h) \ge \pi^i(s, a_h)$$

and since our policies are normalized after every update, $v_{\pi^{i+1}}(s) \ge v_{\pi^i}(s)$. □

## 4 EXPERIMENTS

**Baselines.** For our experiments we compare our method against PPO-DR (Peng et al., 2018) (i.e., PPO trained with domain randomization to improve its robustness), PPO, DQN (Mnih et al., 2013), A2C (Advantage Actor-Critic) (Mnih et al., 2016), when re-trained from scratch on the new world parametrized by $\Delta_{new}$ (except for PPO-DR), as well as when they are trained over the model that produces $\pi^*$ in the world parametrized by $\Delta$. (These are the base models indicated in the figures, and the algorithms are indicated as A2C-B, PPO-B, and DQN-B, respectively.) [2] We use the 'stable-baselines3'(Raffin et al., 2021) implementation for the aforementioned Deep RL algorithms and tuned the hyper-parameters using the 'optuna'(Akiba et al., 2019) library. We test these algorithms on the *FrozenLake*, *CliffWalking*, and *Taxi* environments in Open AI gymnasium, *Minigrid's Distribution Shift* environment (Chevalier-Boisvert et al., 2023) and a version of the *Minigrid: Empty* environment customized with walls and lava. We remark that we use larger and more complex versions of the standard environments to test our algorithm effectively.

1. *FrozenLake:* In the *FrozenLake* environment, the episode terminates when the agent enters a hole or reaches a goal. We have a base model with few holes and three additional levels of environments with increasing grid area occupied by holes, indicated with the darker blue in Fig. 1e.
2. *CliffWalking*: In the *CliffWalking* Environment, if the agent enters a grid location labeled 'cliff', it is returned to the start position. We have a base model with one row of cliffs, similar to the standard model, and three additional levels with increasing cliff area (indicated in brown Fig. 1d). The final level (Level 3) only has a narrow strip across which the agent can walk, making it a very difficult environment to navigate.

---

[2]The experiments were performed on a high-performance computing cluster with nodes using dual 8-16 core processors, using 16 CPUs with 32GB memory each.

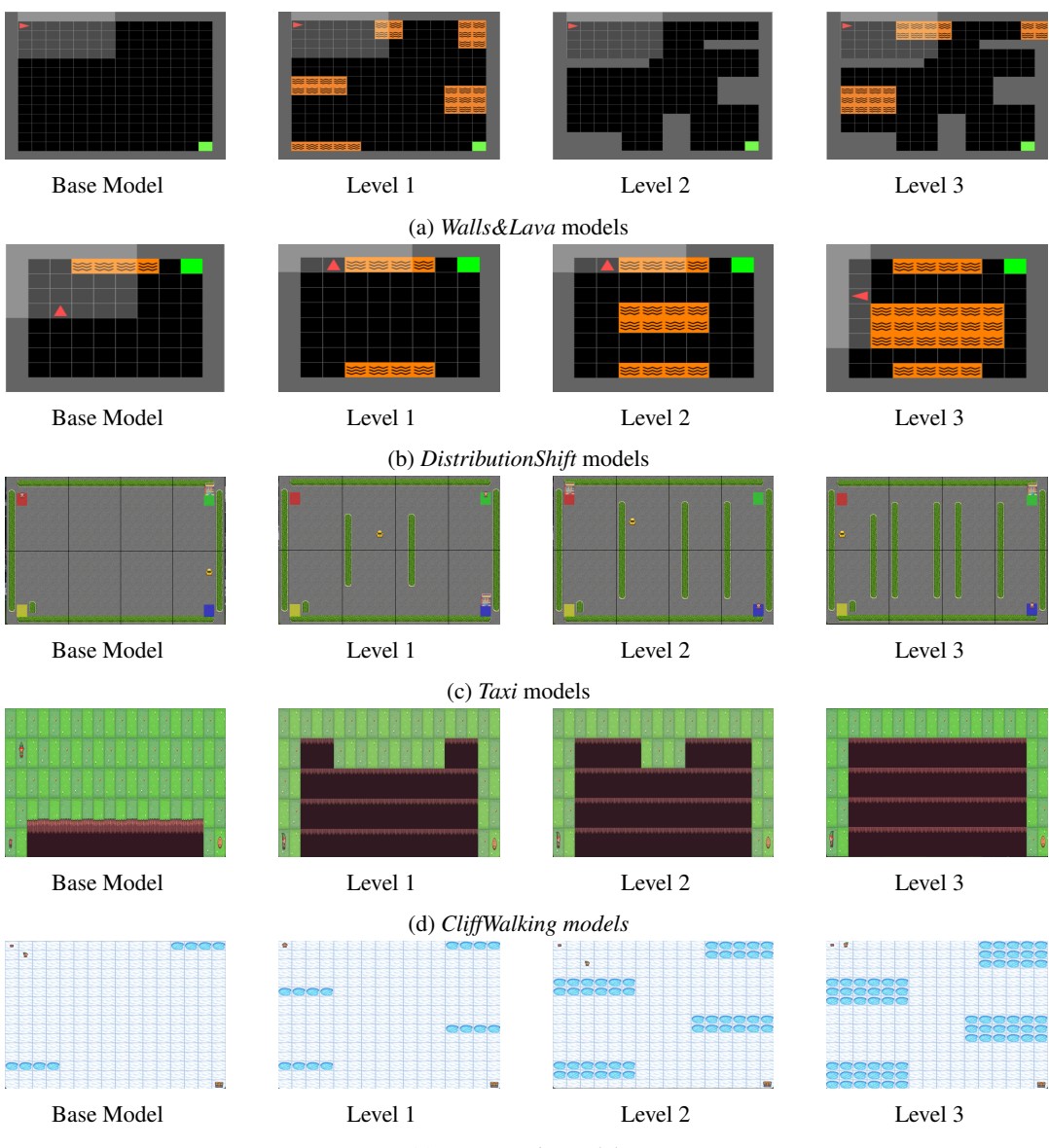

Base Model     Level 1     Level 2     Level 3

(a) *Walls&Lava* models

Base Model     Level 1     Level 2     Level 3

(b) *DistributionShift* models

Base Model     Level 1     Level 2     Level 3

(c) *Taxi* models

Base Model     Level 1     Level 2     Level 3

(d) *CliffWalking models*

Base Model     Level 1     Level 2     Level 3

(e) *FrozenLake* models

(f) Experimental environments including the base model and their variations

3. *Taxi*: A passenger must be picked up from one of the designated stations and dropped off at another. Barriers prevent the Taxi from turning left or right. The base model has no barriers, and additional levels have increasing numbers of barriers (see Fig. 1c), making navigating more circuitous.

4. *Minigrid: DistributionShift*: The aim of this specific environment is to test the ability of the algorithm to generalize across two variations of the environment. The episode terminates when the agent reaches the goal or a location with lava. We have three levels of environments with increasing grid areas occupied by lava. See Fig. 1b.

5. *Minigrid: Walls&Lava*: In this environment, we test the ability of the algorithm to perform navigation in the presence of walls that block the agent's vision and movement, or lava that terminates the episode, or both. The base model is an empty grid, Level 1 has lava, Level 2 has walls, and Level 3 has both. See Fig. 1a.

## 5 RESULTS

We present the results of each environment for ERPO and the other baseline algorithms. In particular, we report the number of timesteps it takes to reach convergence and the actual time elapsed (in seconds). A key feature we observe is that the performance of ERPO does not vary much with increasing levels of difficulty, even when the new environment is drastically different and much more difficult to navigate than the base environment, while all the other algorithms worsen significantly. We also observe that in terms of absolute time taken, ERPO greatly outperforms the others by almost always taking two to three orders of magnitude less time.

From Table 1a, we see that ERPO significantly outperforms the other algorithms with only PPO and A2C approaching convergence but still requiring up to an order of magnitude more timesteps than ERPO in the *Walls&Lava* environment. For the *Minigrid: DistributionShift* environment, we see from Table 1d that A2C-B is similar but slightly worse than ERPO. The other baselines took significantly longer or did not converge. In the results of the *Taxi* environment (see 1c), PPO-DR and PPO-B have the closest performance to the ERPO algorithm. However, the PPO-DR environment requires almost ten times longer, and PPO-B takes up to three times more timesteps. PPO takes five to ten times longer than ERPO to converge. We see that the modified *CliffWalking* environments seem extremely hard to solve (see Table. 1b) because of the large cliff area and episodes that go on almost endlessly (in the original environment there is no termination condition apart from reaching the goal, but we keep an upper limit of 5-10K steps depending on the level). Most algorithms do not converge, but A2C-B performs almost as well as ERPO, albeit taking more absolute time. Finally, in the *FrozenLake* environment, we see that most algorithms perform well albeit requiring 3 to 5 times more timesteps and 10 to 100 times longer than ERPO, but the performance of these algorithms deteriorates significantly with each level while ERPO's performance does not worsen as much (Table 1e).

**Discussion:** We observe that our algorithm outperforms PPO, PPO-DR, and A2C, which rely on batch-wise updates, and DQN which relies on episode-wise updates. These models weight every step equally for performing an update. ERPO, on the other hand, makes sizeable updates for trajectories that either significantly outperform the others in the batch or severely underperform. Most trajectories with fitness close to the average fitness in the batch do not drastically affect the policy update. The fittest trajectories are thus replicated most widely, and within a small number of training episodes, represent most of the batch.

## 6 LIMITATIONS AND FUTURE WORK:

Our set up is limited to discrete state-action spaces. We are working on an extension that works with continuous spaces. This will be carried out with function approximation using radial basis functions that also update the policies of states within a certain distance of the state we are updating. Additionally, because we normalize the probability distribution across actions of a given state, a continuous model would work instead along with a probability density function that can be updated using Dirac delta functions. Our set up is also limited to single agent models (unless extended with independent learning). We are working on extensions that can combine other game-theoretic solution concepts for cooperative multi-agent learning.

## 7 CONCLUSION

We discuss the shortcomings of current deep RL models to adapt to large-scale environmental distribution shifts. To overcome these, we present an algorithm, ERPO, based on the principles of EGT and leverage replicator dynamics to adapt policies from base models for the perturbed environments. Theoretically, we show that our algorithm converges to an optimal policy. Experimentally, we compare our model to various state-of-the-art models in different OpenAI environments and show that our model converges faster in terms of timesteps and absolute time, and yields higher rewards than the others. For future work, we will extend our settings to include continuous state-action spaces and multi-agent environments. Additionally, we will work on providing theoretical guarantees in terms of convergence, and safety and robustness.

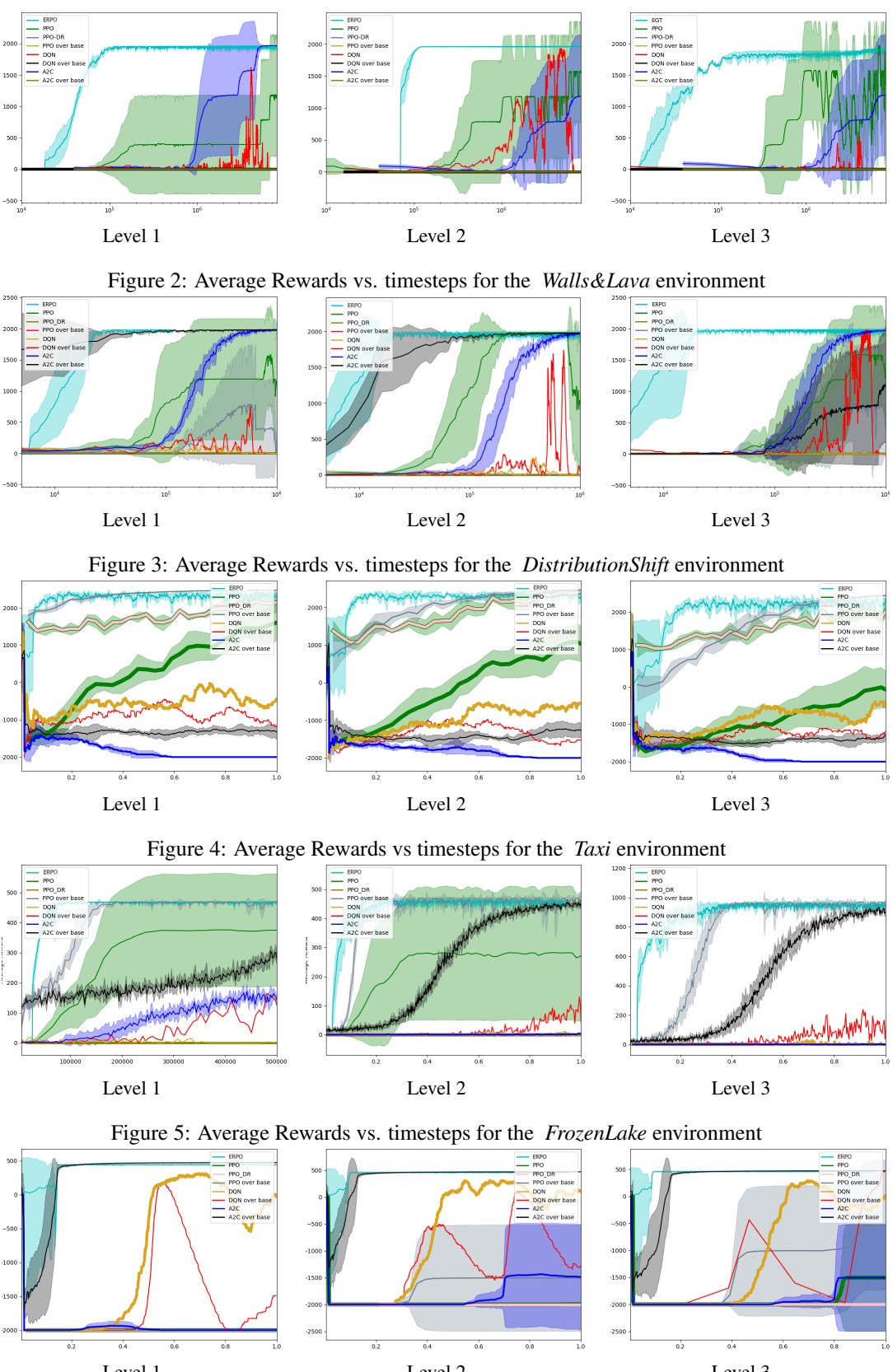

Figure 2: Average Rewards vs. timesteps for the *Walls&Lava* environment

Figure 3: Average Rewards vs. timesteps for the *DistributionShift* environment

Figure 4: Average Rewards vs timesteps for the *Taxi* environment

Figure 5: Average Rewards vs. timesteps for the *FrozenLake* environment

Figure 6: Average Rewards vs. timesteps for the *CliffWalking* environment

| *Walls&Lava* Environment Level | Comparison metric | Algorithms | | | | | | | |
|---|---|---|---|---|---|---|---|---|---|
| | | **ERPO** | PPO | DQN | A2C | PPO-B | DQN-B | A2C-B | PPO-DR |
| Level 1 | Timesteps till convergence ($\times 10^5$) | **1** | 60 | DNC | 50 | DNC | DNC | DNC | DNC |
| | Time elapsed (s) | **50** | 2000 | - | 1500 | - | - | - | - |
| Level 2 | Timesteps till convergence ($\times 10^5$) | **1** | 50 | 50 | 8 | DNC | 50 | DNC | DNC |
| | Time elapsed (s) | **50** | 1500 | 1000 | 1000 | - | 1000 | - | - |
| Level 3 | Timesteps till convergence ($\times 10^5$) | **2** | 50 | 50 | 10 | DNC | DNC | DNC | DNC |
| | Time elapsed (s) | **70** | 1500 | 1000 | 2000 | - | - | - | - |

(a) Results of the *Walls&Lava* Minigrid environment

| *CliffWalking* Environment Level | Comparison metric | Algorithms | | | | | | | |
|---|---|---|---|---|---|---|---|---|---|
| | | **ERPO** | PPO | DQN | A2C | PPO-B | DQN-B | A2C-B | PPO-DR |
| Level 1 | Timesteps till convergence ($\times 10^5$) | **1.8** | DNC | | DNC | DNC | >100 | >1.8 | DNC |
| | Time elapsed (s) | **12** | - | >1000 | - | - | >1000 | 35 | - |
| Level 2 | Timesteps till convergence ($\times 10^5$) | **1.5** | DNC | >100 | >100 | >100 | >100 | >1.8 | DNC |
| | Time elapsed (s) | **20** | - | >1000 | >1000 | >1000 | >1000 | 35 | - |
| Level 3 | Timesteps till convergence ($\times 10^5$) | **1.5** | >100 | >100 | >100 | >100 | >100 | >2 | DNC |
| | Time elapsed (s) | **20** | - | >1000 | >1000 | >1000 | >1000 | 35 | - |

(b) Results of the *CliffWalking* environment

| *Taxi* Environment Level | Comparison metric | Algorithms | | | | | | | |
|---|---|---|---|---|---|---|---|---|---|
| | | **ERPO** | PPO | DQN | A2C | PPO-B | DQN-B | A2C-B | PPO-DR |
| Level 1 | Timesteps till convergence ($\times 10^5$) | **1** | >100 | >500 | >500 | 3 | >100 | >100 | 9 |
| | Time elapsed (s) | **25** | >1500 | >2000 | >2000 | 1250 | >2000 | 250 | 1500 |
| Level 2 | Timesteps till convergence ($\times 10^5$) | **1.8** | >100 | >500 | >500 | 6 | >100 | >100 | 9 |
| | Time elapsed (s) | **45** | >1500 | >2000 | - | 1250 | >2000 | 250 | 1500 |
| Level 3 | Timesteps till convergence ($\times 10^5$) | **2** | >100 | >500 | DNC | 6 | >500 | >100 | 9 |
| | Time elapsed (s) | **50** | >1500 | >2000 | - | 1250 | >2000 | 250 | 1500 |

(c) Results of the *Taxi* environment

| *DistributionShift* Environment Level | Comparison metric | Algorithms | | | | | | | |
|---|---|---|---|---|---|---|---|---|---|
| | | **ERPO** | PPO | DQN | A2C | PPO-B | DQN-B | A2C-B | PPO-DR |
| Level 1 | Timesteps till convergence ($\times 10^4$) | **2** | >500 | >500 | 40 | DNC | >500 | 4 | DNC |
| | Time elapsed (s) | **25** | >1500 | >1500 | 300 | - | 1500 | 35 | - |
| Level 2 | Timesteps till convergence ($\times 10^4$) | **2** | 20 | >500 | 80 | DNC | >500 | 7 | DNC |
| | Time elapsed (s) | **20** | >1500 | >1500 | 300 | - | >1500 | 35 | - |
| Level 3 | Timesteps till convergence ($\times 10^4$) | **2** | >500 | >500 | 40 | DNC | >500 | 100 | DNC |
| | Time elapsed (s) | **10** | >1500 | >1500 | 300 | - | >1500 | 600 | - |

(d) Results of the *DistributionShift* Minigrid environment

| *FrozenLake* Environment Level | Comparison metric | Algorithms | | | | | | | |
|---|---|---|---|---|---|---|---|---|---|
| | | **ERPO** | PPO | DQN | A2C | PPO-B | DQN-B | A2C-B | PPO-DR |
| Level 1 | Timesteps till convergence ($\times 10^4$) | **8** | 25 | 50 | 50 | 15 | >50 | >50 | DNC |
| | Time elapsed (s) | **5** | 300 | 500 | 300 | 150 | 600 | 150 | - |
| Level 2 | Timesteps till convergence ($\times 10^4$) | **25** | 25 | >100 | >100 | 20 | >100 | 90 | DNC |
| | Time elapsed (s) | **15** | 2000 | 800 | 800 | 1500 | 1500 | 200 | - |
| Level 3 | Timesteps till convergence ($\times 10^4$) | **20** | DNC | >100 | >100 | 30 | >100 | 80 | DNC |
| | Time elapsed (s) | **15** | - | 1000 | >2000 | 500 | >2000 | 300 | - |

(e) Results of the *FrozenLake* environment

## 8 REPRODUCIBILITY

We use publicly available environments in the OpenAI gymnasium and compare our algorithm against deep RL models that are available through the 'stable-baselines3' package. We have provided information that can be used to replicate the environments in the Experiments and Results sections. The hyperparameters used in each of these models and implementation details of our algorithm is provided in the Supplementary Material. We believe that these details will be sufficient to reproduce our results. Additionally, we will make our code publicly available upon publication.

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

# A APPENDIX

## A.1 NOTES FOR IMPLEMENTATION:

Below are the default hyperparameters for each of the baselines. Unless specified otherwise in the following section, we will use these.

1. A2C: {policy: MlPPolicy, n_envs : 8, learning_rate: 7e-4, n_steps: 5, gamma: 0.99, gae_lambda: 1.0, ent_coef: 0.0, vf_coef: 0.5, max_grad_norm: 0.5, rms_prop_eps: 1e-5, use_rms_prop: True, use_sde: False, sde_sample_freq: -1, normalize_advantage: False, stats_window_size: 100, verbose: 1, seed: Optional[int] : None, _init_setup_model: True}

2. PPO: { policy: MlPPolicy, n_envs: 8, learning_rate: 0.0003, n_steps: 2048, batch_size: 64, n_epochs: 10, gamma: 0.99, gae_lambda: 0.95, clip_range: 0.2, clip_range_vf: None, normalize_advantage: True, ent_coef: 0.0, vf_coef: 0.5, max_grad_norm: 0.5, use_sde: False, sde_sample_freq: -1, target_kl: None, stats_window_size: 100, tensorboard_log: None, policy_kwargs: None, verbose: 1, seed: None, _init_setup_model: True[source] }

3. DQN: { policy: MlPPolicy, n_envs: 8, learning_rate: 0.0001, buffer_size: 1000000, learning_starts: 50000, batch_size: 32, tau: 1.0, gamma: 0.99, train_freq: 4, gradient_steps: 1, replay_buffer_class: None, replay_buffer_kwargs: None, optimize_memory_usage: False, target_update_interval: 10000, exploration_fraction: 0.1, exploration_initial_eps: 1.0, exploration_final_eps: 0.05, max_grad_norm: 10, stats_window_size: 100, tensorboard_log: None, policy_kwargs: None, verbose: 1, seed: None, _init_setup_model: True }

4. PPO-B: (Same as the PPO settings used for the base model of each environment.)

5. A2C-B: (Same as the A2C settings used for the base model of each environment.)

6. DQN-B: (Same as the DQN settings used for the base model of each environment.)

7. PPO-DR: (Same as the PPO settings used for the base model of each environment.)

- *CliffWalking:*
    1. PPO: { n_steps: 512, learning_rate: 0.00025}
    2. DQN: { exploration_fraction: 0.8, exploration_final_eps: 0.01, learning_starts: 1000 }
    3. A2C: { n_steps: 100, learning_rate: 0.0005}
    4. ERPO: { number_of_episodes_per_batch: 50, w: 0.3, epsilon: 0.01, alpha: 2, nu: 0.05 }

- *DistributionShift:*
    1. PPO: { n_steps: 512, learning_rate: 0.00025}
    2. DQN: { exploration_fraction: 0.8, exploration_final_eps: 0.01, learning_starts: 1000 }
    3. A2C: { n_steps: 500, learning_rate: 0.0005}
    4. ERPO: { number_of_episodes_per_batch: 50, w: 0.3, epsilon: 0.01, alpha: 2, nu: 0.01 }

- *Walls&Lava:*
    1. PPO: { n_steps: 512, learning_rate: 0.00025}

2. DQN: `{ exploration_fraction: 0.7, exploration_final_eps: 0.01, learning_starts: 1000 }`
3. A2C: `{ n_steps: 500, learning_rate: 0.0005}`
4. ERPO: `{ number_of_episodes_per_batch: 30, w: 0.5, epsilon: 0.01, alpha: 2, nu: 0.01 }`

- *Taxi:*
  1. PPO: `{ n_steps: 4096, learning_rate: 0.00025}`
  2. DQN: `{ exploration_fraction: 0.8, exploration_final_eps: 0.01, learning_starts: 1000 }`
  3. A2C: `{ learning_rate: 0.0005}`
  4. ERPO: `{ number_of_episodes_per_batch: 20, w: 0.5, epsilon: 0.01, alpha: 3, nu: 0.01 }`

- *FrozenLake:*
  1. PPO: `{ n_steps: 512, learning_rate: 0.00025}`
  2. DQN: `{ exploration_fraction: 0.8, exploration_final_eps: 0.01, learning_starts: 1000 }`
  3. A2C: `{ n_steps: 250, learning_rate: 0.0005}`
  4. ERPO: `{ number_of_episodes_per_batch: 30, w: 0.2, epsilon: 0.01, alpha: 3, nu: 0.05 }`

