# OpenReview forum: "Robust Policy Optimization with Evolutionary Techniques"
_ICLR.cc/2024/Conference — ICLR 2024 Conference Withdrawn Submission_

### Official Review · Reviewer_uN8Y · 2023-10-30

**Soundness:** 2 fair
**Presentation:** 1 poor
**Contribution:** 2 fair
**Rating:** 3
**Confidence:** 4

**Summary:**

This paper proposed "Evolutionary Robust Policy Optimization (ERPO)", which aims to adapt policies to an altered environment using fewer training steps while getting higher rewards and requiring lower overall computation time.

**Strengths:**

see above

**Weaknesses:**

This paper seems to be an incomplete work, fails to illustrate its motivation and main technical contribution, and misses a lot of important baselines (e.g., from meta-learning literature) in its experiments. Hence, I think it is difficult to evaluate the novelty, effectiveness, and importance based on its current version that needs to be improved significantly.

some comments:

1. I wonder why evolutionary game theory is a proper solution to robust policy optimization.

2. The title is too broad to reflect the main contribution of this work.

3. Please add more baselines and compare ERPO with them in extensive environments in order to evaluate ERPO's soundness.

4. incorrect citation format.

**Questions:**

see above

---

> ### Author Response · Authors · 2023-11-18
> **Response to Reviewer 3**
>
> I wonder why evolutionary game theory is a proper solution to robust policy optimization.
>
> Game-theoretic solutions have long been proposed in the learning domain,  especially in the field of multi-agent learning ([1], [2], [3], [10]). Replicator dynamics, a key feature of evolutionary game theory which also features in our paper, has been used to analyze (reinforcement) learning ([4], [5], [7], [8], [9]). Additionally, we note that they have been used in collision avoidance cases and adapting to environments [6].
> The novelty of our work is to use this approach for policy adaptation in large distribution shifts, an area in which we identified gaps in the current literature. We believe that the theoretical guarantee of convergence to an optimal policy and comparison to seven baselines across five environments solidifies the case for evolutionary game theory to be considered as a solution to robust policy optimization.
>
>       This paper seems to be an incomplete work, fails to illustrate its motivation and main technical contribution
>
> Motivation: As specified in Section 1, the main motivation is the lack of current research in effectively adapting currently optimal policies to perturbed environments. Current Deep-RL and planning methods over-fit to the environment they are trained on and are not able to effectively adapt to large perturbations. Even domain-randomization and other robust learning methods only work for smaller environmental change, and cannot make much use of the learned policy once it is shown to be ineffective in a changed version of the environment
>
> Contributions:  The main technical contribution as specified in the \textbf{Contributions} part in Section 1, is presenting an algorithm that we show theoretically and experimentally to converge to an optimal policy. We also show that our algorithm experimentally outperforms many state-of-the-art algorithms by orders of magnitude in terms of clock-time, number of time-steps, and average reward.
>
>        ...and misses a lot of important baselines (e.g., from meta-learning literature) in its experiments. Hence, I think it is difficult to evaluate the novelty, effectiveness, and importance based on its current version that needs to be improved significantly. Please add more baselines and compare ERPO with them in extensive environments in order to evaluate ERPO's soundness.
>
> We note that we compare with the following seven baselines:
> Algorithms trained from scratch over the new environment: (1) PPO, (2) A2C, (3) DQN;
> Algorithms trained over the base model of the old environment: (4) PPO-B, (5) A2C-B, (6) DQN-B;
> Domain Randomization algorithms: (7) PPO-DR;
> We demonstrate this across five environments, representing nearly all the available gym environments that use discrete state-action spaces. As per our review, we have not encountered forms of meta-learning suitable for such applications. We are open to learn which specific baselines you would like us to additionally consider.
>
>      incorrect citation format
>
> Thank you for pointing this out. We will make the change in the new draft.
>
> References:
>
> [1] Lanctot, Marc, et al. ``A unified game-theoretic approach to multiagent reinforcement learning." Advances in neural information processing systems 30 (2017).
>
> [2] Bloembergen, Daan, et al. ``Evolutionary dynamics of multi-agent learning: A survey." Journal of Artificial Intelligence Research 53 (2015): 659-697.
>
> [3] Jan’t Hoen, Pieter, and Karl Tuyls.``Analyzing multi-agent reinforcement learning using evolutionary dynamics." European Conference on Machine Learning. Berlin, Heidelberg: Springer Berlin Heidelberg, 2004.
>
> [4] Börgers, Tilman, and Rajiv Sarin. ``Learning through reinforcement and replicator dynamics." Journal of economic theory 77.1 (1997): 1-14.
>
> [5] Galstyan, A. (2013). ``Continuous strategy replicator dynamics for multi-agent q-learning." Autonomous agents and multi-agent systems, 26, 37-53.
>
> [6] Hennes, Daniel, Daniel Claes, and Karl Tuyls. ``Evolutionary advantage of reciprocity in collision avoidance."Proc. of the AAMAS 2013 Workshop on Autonomous Robots and Multirobot Systems (ARMS 2013). 2013.
>
> [7] Panozzo, Fabio, Nicola Gatti, and Marcello Restelli. ``Evolutionary dynamics of Q-learning over the sequence form." Proceedings of the AAAI Conference on Artificial Intelligence. Vol. 28. No. 1. 2014.
>
> [8]Ruijgrok, M., and Th W. Ruijgrok. ``Replicator dynamics with mutations for games with a continuous strategy space." arXiv preprint nlin/0505032 (2005).
>
> [9]Tuyls, Karl, et al. ``Extended replicator dynamics as a key to reinforcement learning in multi-agent systems." Machine Learning: ECML 2003: 14th European Conference on Machine Learning, Cavtat-Dubrovnik, Croatia, September 22-26, 2003. Proceedings 14. Springer Berlin Heidelberg, 2003.
>
> [10] Tuyls, Karl, and Simon Parsons. ``What evolutionary game theory tells us about multiagent learning." Artificial Intelligence 171.7 (2007): 406-416.

---

### Official Review · Reviewer_X8NW · 2023-11-01

**Soundness:** 4 excellent
**Presentation:** 4 excellent
**Contribution:** 3 good
**Rating:** 6
**Confidence:** 5

**Summary:**

The manuscript presents a novel method for transfer learning that combines ideas from evolutionary game theory and reinforcement learning. The method operates only with discrete state-action spaces. Overall, the results showcase that the proposed method, ERPO, consistently outperforms several baselines.

**Strengths:**

- The paper is generally well-written, easy to follow and the ideas are conveyed effectively.
- Strong empirical results on several tasks
- Theoretical convergence guarantees

**Weaknesses:**

- No limitations or weaknesses are described in the text
- No hints on how the method can be extended to continuous state-action spaces are given
- The authors mention *"Simulation models are generally simplistic and fail to consider environmental variables ... so they cannot be directly deployed in such applications"*. I have the following issues with the statement:
    - Simulation models are not generally simplistic. Even the less realistic simulators contain quite sophisticated procedures and models behind. Changing "simplistic" to "non-realistic" or just referring to the well-known Sim2Real gap is enough.
    - Most real-world applications are continuous in nature and are quite difficult to discretize or solve with discrete state-action spaces. The manuscript proposes a method that can only operate with state-action spaces. Thus, this sentence is not the best "motivation" for the proposed method.
- The above comment leads to my main "complaint" from the presented work: since most realistic robotic/autonomous systems applications are continuous in nature, how is the proposed method solving the issue it claims to solve? All the experiments as well have nothing to do with robotic applications.

Typos/Minor comments
===================

- Page 2, first paragraph: *"approaches Rajeswaran et al. (2016)train"* -> there is a space missing
- Page 3, Section 3, first paragraph: *"theory (EGT) Smith (1982),Sandholm (2009)"* -> there is a space missing
- Page 3, last sentence: *"As we make state-wise updates, we modify the replicator equation to be We modify this replicator equation as follows:"*

**Questions:**

- What are the main limitations of ERPO?
- How can we extend ERPO to the continuous case?
- How can ERPO be useful in realistic applications of autonomous systems/robotics?

---

> ### Author Response · Authors · 2023-11-19
>
> We thank the reviewer for their comments and address them as follows:
>
>     Weaknesses:
>      No limitations or weaknesses are described in the text
>      No hints on how the method can be extended to continuous state-action spaces are given
>       Questions:
>      What are the main limitations of ERPO?
>     How can we extend ERPO to the continuous case?
>
>
> We will add a limitations and future work section in the updated draft which we also describe here:
>
> Our set up is limited to discrete state-action spaces. We are working on an extension that works with continuous spaces. This will be carried out with function approximation using radial basis functions that also update the policies of states within a certain distance of the state we are updating. Additionally, because we normalize the probability distribution across actions of a given state, a continuous model would work instead along with a probability density function that can be updated using a Dirac delta function. Our set up is also limited to single agent models (unless extended with independent learning). We are working on extensions that can combine other game-theoretic solution concepts for cooperative multi-agent learning.  We also note that ERPO works hand-in-hand with other methods of learning.
>
>     The authors mention "Simulation models are generally simplistic and fail to consider environmental variables ... so they cannot be directly deployed in such applications". I have the following issues with the statement:
>     Simulation models are not generally simplistic. Even the less realistic simulators contain quite sophisticated procedures and models behind. Changing "simplistic" to "non-realistic" or just referring to the well-known Sim2Real gap is enough.
>
> We agree with the reviewer and this will be corrected in the latest draft.
>
>
>
>      Most real-world applications are continuous in nature and are quite difficult to discretize or solve with discrete state-action spaces. The manuscript proposes a method that can only operate with state-action spaces. Thus, this sentence is not the best "motivation" for the proposed method.
>     The above comment leads to my main "complaint" from the presented work: since most realistic robotic/autonomous systems applications are continuous in nature, how is the proposed method solving the issue it claims to solve? All the experiments as well have nothing to do with robotic applications.
>     Questions:
>     How can ERPO be useful in realistic applications of autonomous systems/robotics?
>
>
> This is indeed a valid concern, and we are hoping that the results established in this paper for discrete state/action spaces can extend to realistic environments. However, even within the space of discrete state/action environments, there are some environments that are used in realistic applications. A prominent example is that of the multi-robot warehouse environment that is commonly used as a benchmark for multi-agent path finding (and used in actual package fulfillment centers). We have experiments carried out on single (and multiple) agent versions of this environment, but did not showcase it in the paper as it is designed for multi-agent tasks. We will include some of our results to provide an example of a realistic environment. For complex environments, with large state-action spaces, deep learning/function approximation approaches are suitable tools to supplement ERPO that we will mention in the paper. We will rephrase our introduction and motivation to include generalized reach-avoid/path-planning problems that might undergo distribution shifts.
>
>
>     Typos/Minor comments
>     > Page 2, first paragraph: "approaches Rajeswaran et al. (2016)train" -> there is a space missing
>     > Page 3, Section 3, first paragraph: "theory (EGT) Smith (1982),Sandholm (2009)" -> there is a      space missing
>     > Page 3, last sentence: "As we make state-wise updates, we modify the replicator equation to be   We modify this replicator equation as follows:"
>
> Thank you for pointing out the typos, we will correct them.

---

> > ### Comment · Reviewer_X8NW · 2023-11-22
> >
> > Thank you for the comments/replies. I have no further comments.

---

### Official Review · Reviewer_nzVe · 2023-11-01

**Soundness:** 1 poor
**Presentation:** 1 poor
**Contribution:** 2 fair
**Rating:** 3
**Confidence:** 3

**Summary:**

The paper addresses the problem of adapting reinforcement learning (RL) policies to significant changes in the environment dynamics in robust RL. Many existing methods, like domain randomization and robust policy optimization, fail when test environments differ substantially from training. The authors propose an evolutionary robust policy optimization (ERPO) approach to adapt policies without full retraining. Assuming access to the optimal blackbox policy on the original environment, ERPO explores the new environment using an $\epsilon$-soft version of this policy. It incrementally improves the policy by weighting state-action pairs from fitter trajectories more highly, inspired by evolutionary game theory. Experiments show superior results against methods only trained on the old environment (referred to as base models) or only trained on the new environment. They also compare their algorithm to domain randomization methods such as PPO-DR.

**Strengths:**

The paper's setup tackles an important real-world challenge - adapting black-box RL policies without full retraining. The simplicity and intuitiveness of ERPO are also strengths. Updating actions based on relative expected rewards is an interesting idea. This evolutionary approach avoids needing gradients for the new environment. However, there are some major concerns about the author's implementation of this idea, the theory, and the writing quality and experiments. Specific issues are detailed in the next section recommending rejection.

**Weaknesses:**

The main concern is about the soundness of the algorithm and theoretical claims. Even if we accept the sparse reward assumption, the conclusion of the authors that "*the value of a state can be approximated by the average return across all trajectories containing the state*" is an intuitive statement and needs concrete evidence. Even if we accept this argument, the proof of Theorem 1 is still incorrect. In particular, the proof mixes up the behavior policy $\pi^i\_{train}$ and the learned policy $\pi^i_{new}$ and assumes they have the same sampling distribution, which is clearly not the case. Moreover, I think the current proof, even by fixing all the previous issues, would not work unless we define
$$\pi^{i+1}(a|s) = \pi^i(a|s) \times \frac{\mathbb{E}[f(\tau\_{(s,a)})]}{\mathbb{E}[f(\tau\_{s})]}$$
where, in the denominator, we have $\tau\_{s}$ instead of $\tau\_{s'}$.

Besides the previous concerns, I think the empirical comparison to other methods is unfair. In particular, the proposed method essentially uses the information from the old environment (through the optimal policy) and the data from the new environment. A more fair comparison would initialize the policy of any method that trains on the new environment as the previous policy (e.g., using a cross-entropy loss). The fact that PPO eventually gets to the optimal solution (even without the correct initialization) suggests that initializing its policy with $\pi\_{old}$ will result in comparable or even better results than ERPO.

In summary, concerns include the following:
1. Unjustified approximations in analysis
2. Logical gaps in the convergence proof
3. Unfair comparative evaluations against methods not exploiting old policy information

**Questions:**

Please refer to the previous section.

---

> ### Author Response · Authors · 2023-11-18
> **Response to Reviewer 1 (nzVe)**
>
> The main concern is about the soundness of the algorithm and theoretical claims. Even if we accept the sparse reward assumption, the conclusion of the authors that "the value of a state can be approximated by the average return across all trajectories containing the state" is an intuitive statement and needs concrete evidence.
>
> We follow the definition of Sutton et. al. [1]
> \begin{equation}
>     v_{\pi}(s)= \mathop{\mathbb{E}}[G_t|S_t = s]
> \end{equation}
> where $G_t$ is the return from time $t$ when state $s$ occurs.
> According to our sparse-reward assumption, any reward $r_t \approx 0$ in the transition $(s_t, a_t, r_{t+1}, s_{t+1})$, unless $s_{t+1}$ is the goal/terminal state.
> Therefore:
> \begin{equation}
>     G_t = \sum_{k=t}^{T-1} \gamma^k r^{k+1} = \sum_{k=t}^{T-2} \gamma^k r^{k+1} + \gamma^{T-1} r^{T} \approx \gamma^{T-1} r^{T}
> \end{equation}
> which indicates that $G_t \approx G_0$ for any $t$.
>
> As we have defined:
> \begin{equation}
>     G (\tau) = \sum_{k=0}^{T-1} \gamma^k r^{k+1}
> \end{equation}
>
>
>    From (1), (2), and (3) we can say that
>  $v_{\pi}(s) = \mathop{\mathbb{E}} [G(\tau_s)]$ where $\tau_s = \{ \tau | (s,a) \in \tau, \exists a \in A\}  $
>
>     The proof of Theorem 1 is still incorrect. In particular, the proof mixes up the behavior policy and the learned policy and assumes they have the same sampling distribution, which is clearly not the case.
>
> We remark that using the behavior policy ($\pi_{train}$) for exploration/training while updating the policy we wish to learn $\pi_{new}$ is a technique used in many off-policy methods such as DQN (or any algorithm trained with replay buffers). The convergence of such algorithms does not rely on the sampling distributions of the policies being equivalent.
> Further, we mention that for the $i^{th}$ iteration
>  $\pi^i_{train}$ acts as a $w$-soft version of the learned policy where it assigns the weight $w$ to the old policy and $(1-w)$ to $\pi_{new}^i$ . Therefore, our algorithm continues to train by decrementing  $w$ at each iteration until it is sufficiently small (i.e. $w \leq \epsilon$), by when $\pi_{train}$ is an $\epsilon$-soft version of $\pi_{new}$ .For a sufficiently small $\epsilon$, an $\epsilon$-soft version of $\pi$ converges to optimal if $\pi$ converges to optimal.
>
>      Moreover, I think the current proof, even by fixing all the previous issues, would not work unless we define
>   \\[ \pi^{i+1} = \pi^i(a|s) \times \ \frac{ \mathop{\mathbb{E}}[f(\tau_{(s,a)}]}{\mathop{\mathbf{E}}[f(\tau_{(s)}]} \\]
>
> We agree that this was an oversight on our part, an honest confusion while fleshing out the details. Our experiments have indeed been performed using the aforementioned update rule rather than the erroneous one. We will have corrected this error wherever mentioned in our paper.
>
>     Besides the previous concerns, I think the empirical comparison to other methods is unfair. In particular, the proposed method essentially uses the information from the old environment (through the optimal policy) and the data from the new environment. A more fair comparison would initialize the policy of any method that trains on the new environment as the previous policy (e.g., using a cross-entropy loss). The fact that PPO eventually gets to the optimal solution (even without the correct initialization) suggests that initializing its policy with $\pi_{old}$ will result in comparable or even better results than ERPO.
>
> Please refer to the Baselines paragraph in Section 4 (Experiments), where we mention that we have in fact trained all our algorithms over the old information (base model with the old optimal policy) as well, and denoted them as PPO-B, A2C-B and DQN-B respectively. From tables (a)-(f) we show that these models do outperform PPO, DQN and A2C in some instances such as in the  CliffWalking, Taxi, and  DistributionShift environments as you suggested. However, we can observe that while they perform comparatively at times, they do not outperform ERPO in nearly all of the instances (save in Level 2 of FrozenLake).
>
> Perhaps our framing of these experiments and observations was not explicit enough, and we will do better in adequately highlighting these comparisons. We will re-word our description in these experiments to this effect if it is still unclear.